# Stabilizing Recurrent Dynamics for Test-Time Scalable Latent Reasoning in Looped Language Models

Xiao-Wen Yang [* 1 2]  Zi-Yu Han [* 1 2]  Xi-Hua Zhang [1 2]  Wen-Da Wei [1 2]  Jie-Jing Shao [1]
Lan-Zhe Guo [1 3]  Yu-Feng Li [1 2]

## Abstract

Looped Language Models (LoopLMs) enable efficient latent reasoning through depth recurrence, yet exhibit unreliable test-time scaling behavior: performance often peaks at a certain iteration depth and then collapses with further recurrence. Through latent dynamics analysis, we find an inherent trade-off between stability and effectiveness in existing architectures and strategies. By conceptualizing reasoning as uncertainty reduction, we propose that convergence toward stable fixed points while preserving effectiveness represents a promising way. To this end, we propose **STARS** (STAbility-driven Recurrent Scaling), a training framework that constrains latent states to approach asymptotically stable fixed points. This is realized via efficient Jacobian Spectral Radius Regularization with random loop sampling, enabling STARS to maximize effectiveness while ensuring rigorous stability. Experiments on arithmetic tasks show that STARS achieves reliable test-time scaling, and on complex mathematical reasoning it substantially mitigates performance degradation as recurrence depth increases while also improving peak performance. Code is available at: https://github.com/njuyxw/STARS.

## 1. Introduction

Enhancing the complex reasoning capabilities of Large Language Models (LLMs) through test-time scaling (Zhang et al., 2024; 2025b), which involves allocating additional

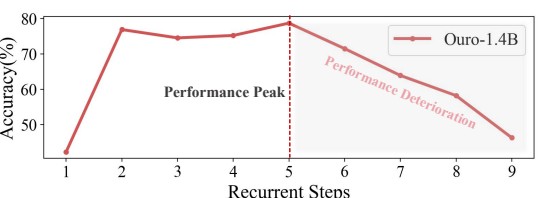

*Figure 1.* Performance of Ouro-1.4B (Zhu et al., 2025b) on GSM8K across different recurrent steps.

computational resources during inference, has become a prominent research focus. The dominant paradigm for test-time scaling relies on generating extensive outputs, typically through chain-of-thought reasoning (Wei et al., 2022) or by sampling multiple candidate solutions and selecting the optimal one (Wang et al., 2022; Yao et al., 2024). Recently, looped language models (LoopLMs) (Geiping et al., 2025; Zhu et al., 2025b) have gained attention as a promising alternative paradigm. By employing depth-recurrence with shared parameters, such models emulate human-like latent reasoning processes (Zhu et al., 2025a). Its advantages include improved computational efficiency, as increased reasoning effort does not entail longer context windows during inference. Moreover, continuous latent representations may offer higher information bandwidth than discrete tokens. Ideally, such architectures should allow for test-time scalable reasoning without expanding the model's parameter count, where increased recurrent iterations lead to progressively refined latent representations (Geiping et al., 2025).

However, our study reveals that if improperly designed, the current LoopLMs often suffer from **unreliable scaling behavior**. Experiments demonstrate that instead of achieving progressive improvements with more computation, performance often exhibits a peak at a certain iteration depth and deteriorates sharply or even collapses entirely as iterations further increase (Figure 1). This phenomenon indicates that direct supervised fine-tuning fails to equip the model with a test-time scalable reasoning capability through depth recurrence. Instead, the model tends to overfit to the specific recurrent iteration during training.

To demystify these scaling failures, we pivot to a dynamical systems perspective to conduct a systematic diagnostic study of LoopLM's latent trajectories. This analysis allows

[1]State Key Laboratory of Novel Software Technology, Nanjing University, Nanjing, China [2]School of Artificial Intelligence, Nanjing University, Nanjing, China [3]School of Intelligence Science and Technology, Nanjing University, Nanjing, China. Correspondence to: Yu-Feng Li <liyf@nju.edu.cn>.

*Proceedings of the 43rd International Conference on Machine Learning*, Seoul, South Korea. PMLR 306, 2026. Copyright 2026 by the author(s).

us to uncover a fundamental, yet previously overlooked, irreconcilable trade-off between effectiveness and stability in current designs. We find that the stability of the latent trajectory is largely determined by where normalization is placed. Internal normalization (e.g., Pre-Norm) maintains information flow (effectiveness) but causes hidden states to grow exponentially, leading to trajectory divergence. Conversely, External normalization (e.g., Post-Norm) ensures bounded states (stability) but often fails to perform deep reasoning. Our experiments show that common remedies, such as auxiliary Prelude/Coda layers, L2 regularization, or random loop sampling, cannot totally resolve this deadlock.

A key insight of this paper is that test-time scalable latent reasoning must satisfy both effectiveness and stability. We argue that reasoning is an iterative process of reducing uncertainty and refining thoughts. In terms of dynamics, this means the hidden states should converge toward an effective and stable fix point. If a system is effective but unstable, the thoughts become chaotic; if it is stable but ineffective, the thoughts remain shallow. To achieve this, we propose **STARS** (STAbility-driven Recurrent Scaling), a unified training framework that integrates Jacobian Spectral Radius Regularization (JSRR) with random loop sampling. According to the Lyapunov Linearization Theorem, the stability of a nonlinear system is determined by the spectral radius of its Jacobian matrix. STARS mathematically compels the model to converge toward asymptotically stable fixed points by constraining this spectral radius during training. To ensure that the framework is practical for large-scale LLMs, we avoid the prohibitive cost of direct eigenvalue calculation. Instead, STARS employs a lightweight and efficient estimation scheme by combining single-step power iteration with Jacobian-vector products (JVP). By applying random loop sampling, STARS optimizes both effectiveness and stability of latent trajectories on a global scale, enabling more robust test-time scaling.

We evaluate the effectiveness of our proposed method through two experimental setups: basic arithmetic tasks on randomly initialized Transformers (Vaswani et al., 2017) and fine-tuning pre-trained LoopLM on complex mathematical reasoning tasks. Results show that on arithmetic tasks, our method achieves fully reliable test-time scaling. On mathematical reasoning tasks, it demonstrates more robust performance compared to baselines. For instance, on GSM8K, while Ouro-1.4B experiences a 20.47% drop from its peak performance after 8 recurrent steps, our method degrades by only 8.26%. Moreover, our approach improves peak performance by 4.01% on GSM8K at the same time.

## 2. Related Work

**Test time scaling and latent reasoning.** Test-time compute scaling is a critical frontier for enhancing LLM rea-

soning (Zhang et al., 2025b). Conventional approaches primarily rely on explicit reasoning, where models generate natural language intermediate steps such as Chain-of-Thought prompting (Wei et al., 2022) to solve complex tasks. Further gains have been achieved through search-based strategies, including majority voting (Wang et al., 2022), Tree-of-Thoughts (ToT) (Yao et al., 2024), and Monte Carlo Tree Search (MCTS) (Yang et al., 2022), which allow the exploration of multiple reasoning paths. However, these methods remain inherently constrained by the bandwidth and efficiency of natural language generation. Inspired by the human tendency to reason through internal steps rather than producing immediate verbal outputs (Zelikman et al., 2024), recent work has pursued latent reasoning, which shifts computation from discrete tokens into latent representations (Zhu et al., 2025a). This approach is more computationally efficient and better suited for abstract reasoning. Coconut (Hao et al., 2024) employing continuous thought tokens derived from previous hidden states is a typical technique. Approaches such as SIM-CoT (Wei et al., 2025) and others (Mohtashami et al., 2023; Shen et al., 2025; Cheng & Van Durme, 2024; Zhang et al., 2025a) share similar ideas. Nevertheless, these approaches essentially remain forms of test-time scaling along the sequential dimension.

**Recurrent Transformers and LoopLM.** Unlike methods that scale sequence length, a newer paradigm of latent reasoning focuses on scaling model depth through recurrence and parameter sharing. Universal Transformer (Dehghani et al., 2018) pioneered dynamic recurrence across layers, establishing depth-adaptive computation as an alternative to fixed-depth transformers. Subsequent research (Giannou et al., 2023; Yang et al., 2023; Fan et al., 2024) have investigated their potential benefits through theoretical and small-scale empirical analyses. Recent studies (Geiping et al., 2025; Zhu et al., 2025b; Du et al., 2025; McLeish et al., 2025) have extended the recurrent Transformer architecture into language models. Huginn (Geiping et al., 2025) trained a 3.5B model from scratch, while Ouro (Zhu et al., 2025b) introduced a more capable LoopLM, enabling it to compete with mainstream open-source LLMs. As a typical latent reasoning approach, LoopLM is expected to demonstrate test-time scaling. However, we find that rather than producing progressive gains with increased computation, performance typically peaks at a specific iteration depth and declines sharply beyond it. This phenomenon is especially pronounced in Ouro models. Existing studies lack a thorough analysis of LoopLM training design, particularly regarding latent dynamics. While DEQ (Bai et al., 2019; 2021) examined the dynamics of looped networks, their architectures and training algorithms differ from standard language models. Therefore, investigating the latent dynamics in modern LoopLMs is crucial to address their unreliable scaling behavior.

## 3. Preliminaries

### 3.1. Looped Language Models

In contrast to traditional deep architectures that stack distinct layers, a looped language model leverages weight-sharing by iteratively applying a recurrent block $\mathcal{M}^L$. The architecture is defined as:

$$\mathcal{F}^{(t)}(\cdot) = \text{lmhead} \circ \text{coda} \circ \underbrace{\mathcal{M}^L \circ \cdots \circ \mathcal{M}^L}_{t \text{ iterations}} \circ \text{prelude}(\cdot),$$

where $\circ$ denotes function composition and,

- prelude : $\mathbb{R}^{M \times |V|} \to \mathbb{R}^{M \times d}$ maps a sequence of $M$ tokens to $d$-dimensional embeddings as a preprocess of the input text.
- $\mathcal{M}^L$ : $\mathbb{R}^{M \times d} \to \mathbb{R}^{M \times d}$ denotes a stack of $L$ causal transformer layers ($\mathcal{T}_{\theta_L} \circ \cdots \circ \mathcal{T}_{\theta_1}$) with hidden size $d$.
- coda : $\mathbb{R}^{M \times d} \to \mathbb{R}^d$ transforms the final iterative representations for the output layer.
- lmhead : $\mathbb{R}^d \to \mathbb{R}^{|V|}$ projects the output back to the vocabulary of size $V$ for generation.

For a special case where $t = 1$, the architecture reduces to a standard non-looped model $\mathcal{F}^{(1)} \equiv F$. For a training batch $\mathcal{D} = \{\mathbf{x}^{(i)}\}_{i=1}^N$, we define the standard cross-entropy loss at $t$ iterations as:

$$\mathcal{L}_{\text{SFT}}^{(t)} = \frac{1}{N} \sum_{i=1}^{N} \sum_{\ell=1}^{M_i - 1} - \log p_\theta^{(t)} \big( x_{\ell+1}^{(i)} \mid x_{1:\ell}^{(i)} \big) \qquad (1)$$

where the conditional probability is given by $p_\theta^{(t)}(\cdot \mid x_{1:\ell}) = \text{softmax}(\text{lmhead}(h_\ell^{(t)}))$. Here, $x_{1:\ell}$ represents the prefix of length $\ell$, and $h_\ell^{(t)}$ denotes the hidden state at position $\ell$ after $t$ recursive iterations.

### 3.2. Latent Reasoning as a Dynamical System

The iterative computation in LoopLM naturally defines a discrete-time dynamical system in latent space. For a given input $\mathbf{x}$, after initial embedding, the recurrent transformation yields the evolution

$$\mathbf{h}^{(t+1)} = \Phi_\theta(\mathbf{h}^{(t)}), \quad t = 0, 1, 2, \ldots, \qquad (2)$$

where $\mathbf{h}^{(t)} \in \mathbb{R}^D$ ($D = M \cdot d$) denotes the flattened latent state at iteration $t$ and $\Phi_\theta := \mathcal{M}^L$ is the deterministic nonlinear map parametrized by $\theta$. The trajectory is the sequence $(\mathbf{h}^{(0)}, \mathbf{h}^{(1)}, \ldots)$. From this perspective, test-time scaling corresponds to extending the trajectory length of the system, increasing the number of iterations $t$ without changing parameters. The success of such scaling therefore depends on the long-term behavior of the trajectory.

**Attractor and fixed points.** In dynamical systems theory, an attractor describes the long-term evolution of a system.

Formally, a set $\mathcal{A} \subset \mathbb{R}^D$ is an attractor for the dynamics induced by $\Phi_\theta$ if there exists a neighbourhood $\mathcal{U} \supset \mathcal{A}$ such that, for every state $\mathbf{h}^{(t)} \in \mathcal{U}$, $\lim_{t \to \infty} \text{dist}(\mathbf{h}^{(t)}, \mathcal{A}) = 0$, where $\text{dist}(\cdot, \mathcal{A})$ denotes the distance to $\mathcal{A}$. Intuitively, attractors are regions of latent space toward which the model's internal representations evolve during reasoning. A fixed point is a particularly important type of attractor. A state $\mathbf{h}^\star \in \mathbb{R}^D$ is a fixed point if $\Phi_\theta(\mathbf{h}^\star) = \mathbf{h}^\star$. Once the latent trajectory reaches such a state, further applications of the recurrent block leave the representation unchanged. In LoopLM, fixed points correspond to stable internal representations that signify the completion of the reasoning process. A fixed point $\mathbf{h}^\star$ is locally asymptotically stable if there exists $\epsilon > 0$ such that for every $\mathbf{h}^{(t)}$ with $\|\mathbf{h}^{(t)} - \mathbf{h}^\star\| < \epsilon$, $\lim_{t \to \infty} \mathbf{h}^{(t)} = \mathbf{h}^\star$. Such stable fixed points are desirable for reasoning tasks, as they guarantee convergence rather than oscillation or divergence when inference is extended.

## 4. Recurrent Dynamics of LoopLM

To delve into the core factors governing the intrinsic dynamics of LoopLM, this section presents a series of systematic diagnostic experiments. Section 4.1 first outlines the specific experimental setup. Subsequently, Section 4.2 analyzes the impact of critical architectural components on reasoning instability in recurrent models. Finally, Section 4.3 explores potential strategies for scalable latent reasoning.

### 4.1. Experimental Setup

Our study is grounded in a controlled environment designed to make latent dynamics observable. We utilize multi-digit addition as an idealized testbed for analyzing iterative algorithmic reasoning.

**Task description.** The models are trained on 4-digit by 4-digit addition problems (e.g., "1234 + 5678 = 6912") using a vocabulary consisting of digits, arithmetic operators, and special tokens. The training dataset comprises 100,000 samples which are randomly generated, a scale sufficient to preclude rote memorization and compel the model to learn a generalized addition algorithm. The choice of 4-digit addition balances computational constraints with task complexity: while 2-digit addition offers a trivial sample space ($10^4$ combinations), 4-digit addition presents a vastly larger space ($10^8$ combinations), providing a sufficiently complex landscape for analysis and posing a non-trivial challenge for non-pretrained Transformers.

**Model architecture.** We employ a standard GPT-style Transformer block (Vaswani et al., 2017; Radford et al., 2019) as a minimal recurrent unit ($L = 1$), realizing the looped model through the recursive iteration of this unit. This architectural choice eliminates confounding variables

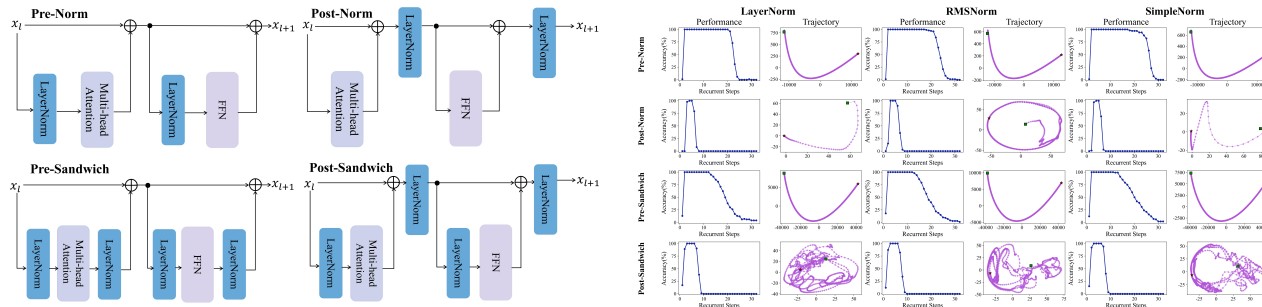

*Figure 2.* Left: Structural Diagrams. Right: Visualization showing accuracy evolution and latent state dynamics.

associated with deep, heterogeneous stacked layers, enabling us to isolate performance variations as direct consequences of iteratively applying a single, well-defined state transition function. The unit hyperparameters are set to $d_{\text{model}} = 512$, $n_{\text{heads}} = 8$, and $d_{\text{ff}} = 1024$.

**Evaluation details.** For the static architecture analysis, models are trained with a fixed loop iteration count of $T_{\text{train}} = 4$. During evaluation, we sweep the test-time iteration count ($T_{\text{test}}$) across a broad spectrum. Our primary focus is to characterize the evolution of system stability and internal state trajectories when $T_{\text{test}}$ diverges from, and specifically exceeds, the training horizon $T_{\text{train}}$. We then perform PCA on the sequence of latent states and project them onto a two-dimensional space spanned by the first two principal components for visualization.

### 4.2. Static Architecture Analysis

#### 4.2.1. IMPACTS OF NORM STRUCTURE DESIGN

The architectural configuration of the recurrent unit fundamentally dictates the evolution of information flow. We conduct a comparative analysis of three normalization variants Layer Normalization (Ba et al., 2016), RMSNorm (Zhang & Sennrich, 2019), and SimpleNorm (defined as normalization without learnable affine parameters) across four structural positions. Our choice of normalization placement: Pre-, Post-, Pre-Sandwich, and Post-Sandwich (see Figure 2), is primarily motivated by its substantial impact on training stability and model performance, as established in prior research (Xiong et al., 2020; Bai et al., 2021). This yields a total of twelve distinct architecture combinations.

Based on these formulations, we categorize the architectures into two distinct system types: *internal normalization system*, where the residual connection remains outside the normalization scope (Pre-Norm and Pre-SandwichNorm), and *external normalization system*, where the residual stream is encapsulated within the final normalization scope (Post-Norm and Post-SandwichNorm).

Our experiments reveal an inherent design dilemma within recurrent Transformer architectures. As visualized in Figure 2, the specific choice of normalization operator exerts minimal influence on the model's latent dynamics; conversely, the structural placement of the normalization layer is the determining factor, precipitating two distinct failure modes.

> **Finding 1:** The specific normalization operator has negligible impact on model dynamics, whereas the structural placement of the layer dictates the system's evolutionary behavior.

In the *internal normalization system*, we observe that while performance is maintained throughout the training horizon and brief subsequent extrapolations, it deteriorates as test iterations increase. The PCA trajectories exhibit massive scaling, indicating that the hidden states gradually drift away from the effective structural manifold, resulting in performance collapse. Specifically, in the Pre-Norm formulation ($x_{l+1} = x_l + f(\text{Norm}(x_l))$), the residual connection establishes an information highway (He et al., 2016), transmitting the previous state to the next timestamp without attenuation. However, the update vector $f(\text{Norm}(x_l))$ is directly accumulated onto the backbone stream. Without a constraint mechanism after the residual addition, the norm of the hidden state tends to grow without bound. This creates a positive feedback loop where the state magnitude amplifies linearly with iterations, eventually diverging from the functional data manifold. Conversely, in *external normalization system*, although the model sustains performance for only a short duration during testing, the latent dynamics remain bounded, as evidenced by the compact scale of the PCA projections. As test iterations progress, both system types eventually converge towards attractor. Consequently, current normalization architectures impose a trade-off between system stability and effective information propagation, with neither paradigm achieving test-time scalable latent reasoning.

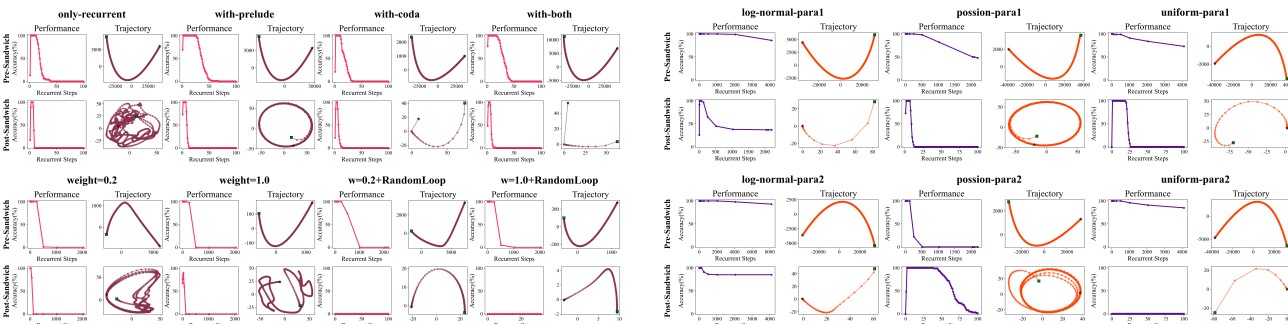

*Figure 3.* Left: The top panel analyzes the impact of adding non-recurrent layers, while the bottom panel assesses the effect of introducing L2 regularization. Right: This panel evaluates the random loop strategy across distributions and parameter sets (detailed in Table 1).

> **Finding 2:** *Internal normalization system* allows strong information to flow through the residual backbone. This maintains early performance, but can lead to instability and divergent states. *External normalization system* constrains the residual stream, which ensures stable boundaries, but causes the system to settle into poor states that degrade its performance.

### 4.2.2. IMPACTS OF PRELUDE AND CODA LAYERS

A natural consideration is whether incorporating non-recurrent layers before and after the recurrent block, similar to the approach in Huginn (Geiping et al., 2025), can alleviate these limitations. To investigate this, we add Prelude and Coda layers into our experiments. The added Prelude and Coda layers utilize the same block architecture as the recurrent unit but do not participate in the loop iterations.

Given our previous finding that the specific normalization type has negligible impact, we fix LayerNorm as the normalization operator for this analysis. We conduct experiments using two structural baselines Pre-Sandwich and Post-Sandwich across four configurations: *only-recurrent*, *with-prelude*, *with-coda*, and *with-both*.

As visualized in Figure 3, for the *internal normalization system*, the addition of a prelude layer marginally slows performance degradation, but the effect is negligible; the trajectory's drift scale remains immense, and accuracy rapidly collapses to zero. For the *external normalization system*, while a coda layer leads to a more concentrated set of final attractor, these prove to be non-benign fixed points that offer no performance benefit.

> **Finding 3:** The inclusion of Prelude and Coda layers does not alter the system's dynamical tendencies. Although these layers likely contribute to a deeper semantic understanding of the input and output spaces, their impact on the system's stability is negligible.

*Table 1.* Hyperparameter settings for random loop distributions. Range indicates the clipping bounds [min, max].

| Distribution | Set 1 Configuration | Set 2 Configuration |
|---|---|---|
| Log-Normal | $\mu = 2.62, \sigma = 0.60$ range: $[1, 40]$ | $\mu = 2.00, \sigma = 0.70$ range: $[1, 100]$ |
| Poisson | $\lambda = 5$ range: $[1, 30]$ | $\lambda = 10$ range: $[1, 30]$ |
| Uniform | range: $[1, 10]$ | range: $[1, 40]$ |

### 4.3. Attempted Methods for Scalable Latent Reasoning

#### 4.3.1. RANDOM LOOP SAMPLING

The random loop sampling strategy aims to decouple model performance from a fixed training step count $T_{\text{train}}$, by dynamically sampling the number of recurrent iterations for each batch. This approach was previously employed in training recurrent models (Geiping et al., 2025); however, a systematic analysis of its impact and insights into its underlying mechanics were not provided. To address this gap, we build upon this method by conducting a detailed investigation. Specifically, we experiment with three distinct distributions (Log-Normal, Poisson, and Uniform), each with two different parameter configurations, applied to our two representative architectures: Pre-Sandwich and Post-Sandwich (others are detailed in Appendix C).

As illustrated in the Figure 3, sampling iterations during training contributes to performance retention. For Pre-Sandwich models, the random loop sampling strategy proves highly effective. Across all combinations, performance retention persists far beyond the iteration range encountered during training. However, this does not alter the fundamental nature of *external normalization system*, and state drift still occurs. For Post-Sandwich models, compared to the trajectory plots in previous experiments, the set of attractors it eventually converges to is more compact, but the training process is unstable, and sometimes the model even fails to learn to solve the task. Furthermore, based on the log-

normal and uniform combinations, a wider sampling range and a higher mean are not necessarily better. For example, when encountering a wider sampling range, Post-Sandwich models are prone to training collapse and fail to learn the task. However, the Post-Sandwich architecture holds greater potential due to its inherent tendency to converge to an attractor. If this convergence can be guided toward an effective attractor, the system could simultaneously achieve both stability and effectiveness.

> **Finding 4:** Overall, sampling loop counts during training significantly enhances performance retention and generalization capabilities. However, it does not fundamentally transform the effective stability of the system and is sensitive to the size of the sampling range.

### 4.3.2. L2 REGULARIZATION

To curb the continuous drift in *internal normalization system* and enhance the effective stability of *external normalization system*, a natural idea is to introduce a regularization term defined as the $L_2$ norm of the difference between hidden states across adjacent iterations. However, as depicted in the lower panel of the Figure 3, the impact of L2 regularization is minimal. It provides only marginal improvements in performance retention and drift mitigation for *internal normalization system*, while its effect on *external normalization system* is virtually non-existent. Furthermore, we evaluated the combination of the random loop sampling strategy with L2 regularization. Contrary to expectations, this combination yields no enhancement and actually underperforms compared to the pure random loop sampling.

> **Finding 5:** The introduction of an L2 regularization between hidden states yields negligible benefits and demonstrates poor synergy when combined with the random loop sampling strategy.

## 5. Jacobian Spectral Radius Regularization

Building on our previous experimental findings, we propose that effective test-time scalable latent reasoning must satisfy both effectiveness and stability. This requirement stems from the nature of reasoning as an iterative process of uncertainty reduction and thought refinement. This implies that hidden states should converge toward a stable and effective fixed point. Systems that are effective but unstable produce chaotic reasoning trajectories, whereas systems that are stable but ineffective yield shallow thoughts. To address this, we introduce STARS (STAbility-driven Recurrent Scaling), a training framework that combines Jacobian Spectral Radius Regularization (JSRR) with random loop sampling.

According to the Lyapunov Linearization Theorem for discrete-time dynamical systems, the local stability of a

nonlinear system defined by $\mathbf{h}^{(t+1)} = \Phi_\theta(\mathbf{h}^{(t)})$ at a fixed point $\mathbf{h}^\star$ is governed by the properties of its Jacobian matrix, denoted as:

$$J(\mathbf{h}^\star) = \nabla_\mathbf{h} \Phi_\theta(\mathbf{h})|_{\mathbf{h}=\mathbf{h}^\star}$$

To determine stability, we examine the set of eigenvalues $\{\lambda_1, \lambda_2, \ldots, \lambda_n\}$ of $J(\mathbf{h}^\star)$. The critical metric is the spectral radius, denoted by $\rho(J(\mathbf{h}^\star))$, which is defined as the maximum absolute value (modulus) of these eigenvalues:

$$\rho(J(\mathbf{h}^\star)) = \max_i\{|\lambda_i|\}$$

Specifically, if the spectral radius satisfies: $\rho(J(\mathbf{h}^\star)) < 1$, the fixed point is *asymptotically stable*. Under this condition, any small perturbations introduced to the system will decay exponentially over successive iterations. And if the spectral radius is smaller, the convergence rate also becomes faster.

Previous works (Bai et al., 2019; 2021) often regularize using the Frobenius norm $\|J\|_F$. However, while the spectral radius and the norm satisfy $\rho(J) \leq \|J\|$, directly constraining the norm is overly restrictive and may excessively compress the model's expressive capacity. In contrast, regularizing the spectral radius provides a mathematically precise means to achieve stability.

Direct eigenvalue computation of $J \in \mathbb{R}^{D \times D}$ is infeasible for large $D$. We therefore adopt an efficient spectral radius estimator based on the power iteration method. Starting from a randomly initialized vector $\mathbf{v}$, the power iteration procedure repeatedly updates $\mathbf{v} \leftarrow \frac{J\mathbf{v}}{\|J\mathbf{v}\|}$, eventually converging to the dominant eigenvector of $J$. The corresponding spectral radius can then be estimated by: $\rho(J) \approx \|J\mathbf{v}\|_2$. To integrate this approach into large-scale model training, we use only a single-step power iteration with the Jacobian-vector product technique in Pytorch, enabling efficient and memory-aware computation without explicit construction of the full Jacobian matrix.

We adopt a single-step update for two reasons: 1) multi-step iteration introduces complex gradient dependencies that can lead to abnormal gradients; 2) It is computationally lightweight, and experiments show it provides effective supervisory signals. Although the single-step estimate may be noisy for individual samples, its optimization direction remains statistically accurate across batches.

While the ultimate goal is to optimize the spectral radius at the fixed point $\mathbf{h}^\star$, identifying the exact location of the fixed point during training is computationally prohibitive. Consequently, we adopt a proxy approach: for a specific iteration $t$ within the LoopLM execution, we regulate the squared spectral radius of the Jacobian at the current state $\mathbf{h}^{(t)}$. For a batch of $N$ samples, the JSRR loss at iteration $t$ is formulated as:

$$\mathcal{L}_{\text{JSRR}}^{(t)} = \frac{1}{N} \sum_{i=1}^{N} \left\| J^{(t,i)} \mathbf{v}^{(t,i)} \right\|_2^2, \tag{3}$$

*Table 2.* We report Accuracy (%) on various mathematical benchmarks. Best results for LoopLMs are **bolded**.

| Model | Recurrents | GSM8K | MATH500 | ASDiv | SVAMP | AMC23 | AVG |
|---|---|---|---|---|---|---|---|
| *Standard Language Models* | | | | | | | |
| Gemma3-1b-it (Team et al., 2025) | - | 42.76 | 41.20 | 72.45 | 62.67 | 25.00 | 48.82 |
| Llama-3.2-3B-Instruct (Grattafiori et al., 2024) | - | 71.11 | 42.40 | 82.04 | 79.67 | 15.00 | 58.04 |
| Qwen2.5-3B-Instruct (Team et al., 2024) | - | 74.91 | 68.20 | 76.18 | 87.00 | 32.50 | 67.76 |
| Qwen3-1.7B (Yang et al., 2025) | - | 75.82 | 65.80 | 85.64 | 89.00 | 17.50 | 66.75 |
| Qwen3-4B (Yang et al., 2025) | - | 80.36 | 63.00 | 86.59 | 89.00 | 12.50 | 66.29 |
| *Looped Language models* | | | | | | | |
| | 2 | 15.01 | 14.60 | 30.50 | 18.67 | 00.00 | 15.76 |
| Recurrent-Llama-3.2 (McLeish et al., 2025) | 4 | 14.03 | 14.60 | 41.65 | 18.67 | 5.000 | 18.79 |
| | 8 | 13.04 | 14.00 | 21.33 | 42.17 | 7.500 | 19.61 |
| | 2 | 17.89 | 9.000 | 29.33 | 31.97 | 00.00 | 17.64 |
| Recurrent-OLMo-2-0425 (McLeish et al., 2025) | 4 | 17.13 | 9.800 | 38.13 | 38.67 | 2.500 | 21.25 |
| | 8 | 17.97 | 11.40 | 40.00 | 38.48 | 00.00 | 21.57 |
| | 2 | 76.88 | 54.80 | 74.27 | 81.67 | 35.00 | 64.52 |
| Ouro-1.4B (Zhu et al., 2025b) | 4 | 75.21 | 59.60 | 76.57 | 75.67 | 50.00 | 67.41 |
| | 8 | 58.23 | 40.80 | 70.07 | 66.33 | 40.00 | 55.09 |
| | 2 | 76.04 | 55.80 | 82.08 | 82.33 | 30.00 | 65.25 |
| Ouro-1.4B-SFT | 4 | 80.06 | 64.60 | 83.47 | 76.67 | 47.50 | 70.46 |
| | 8 | 60.05 | 39.20 | 75.10 | 68.00 | 22.50 | 52.97 |
| | 2 | 75.89 | 56.40 | 82.56 | 79.67 | 40.00 | 66.90 |
| Ouro-1.4B-STARS (Ours) | 4 | **81.96** | **67.40** | **84.73** | **84.33** | **52.50** | **74.18** |
| | 8 | 74.45 | 54.80 | 82.52 | 81.00 | 35.00 | 65.55 |

where $J^{(t,i)} = \frac{\partial \mathbf{h}_i^{(t+1)}}{\partial \mathbf{h}_i^{(t)}}$ is the Jacobian of the $i$-th sample, and $\mathbf{v}_1^{(t,i)}$ is the one-step dominant eigenvector estimate of the $i$-th sample at iteration $t$.

However, directly constraining the spectral radius at only a single iteration $t$ is suboptimal, as it neglects the stability properties across the entire inference trajectory. Therefore, we propose to integrate JSRR with the random loop sampling strategy introduced earlier. This ensures that spectral radius regularization is applied across a diverse set of states encountered during iterative inference, promoting global stability over the support set of the latent dynamics. Our final training objective is defined as the expectation of the combined loss over the recurrent depth distribution $\mathcal{P}$:

$$\mathcal{L}_{\text{STARS}} = \mathbb{E}_{t \sim \mathcal{P}} \left[ (1 - \lambda) \cdot \mathcal{L}_{\text{SFT}}^{(t)} + \lambda \cdot \mathcal{L}_{\text{JSRR}}^{(t)} \right], \quad (4)$$

where $\lambda$ is a balancing hyperparameter. By sampling the loop length $t$ from the distribution $\mathcal{P}$, this formulation encourages the model to achieve high performance on the training data while maintaining a small spectral radius across the entire trajectory. The complete procedure of our algorithm is presented in Algorithm 1.

## 6. Experiments

In this section, we conduct experiments to demonstrate the effectiveness of our proposed method. We mainly focus on two tasks: the arithmetic task mentioned in Section 4 and complex mathematical reasoning benchmarks of a pre-trained LoopLM model, exemplified by Ouro. Furthermore, we conduct rigorous ablation and analysis studies.

### 6.1. Experimental Setup

**Multi-digits addition task.** We follow the experimental setup described in Section 4 and adopt a Post-Sandwich LayerNorm structure due to our previous findings. For the random loop configurations, we employ log-normal random loop sampling ($\mu = 2, \sigma = 0.7, \text{range} = [1, 100]$) with a weight $\lambda = 0.1$. The training is conducted with a learning rate of $1 \times 10^{-4}$.

**Mathematical reasoning task.** We employ Ouro-1.4B (Zhu et al., 2025b) which is a pre-trained LoopLM as our base model, specifically targeting its unreliable scaling behavior in test-time scaling. We fine-tuned the model on a random subset of NuminaMath-1.5 (Li et al., 2024) dataset containing 400K samples due to the computational resource constraints. The training configuration involved log-normal random loop sampling ($\mu = 1.7, \sigma =$

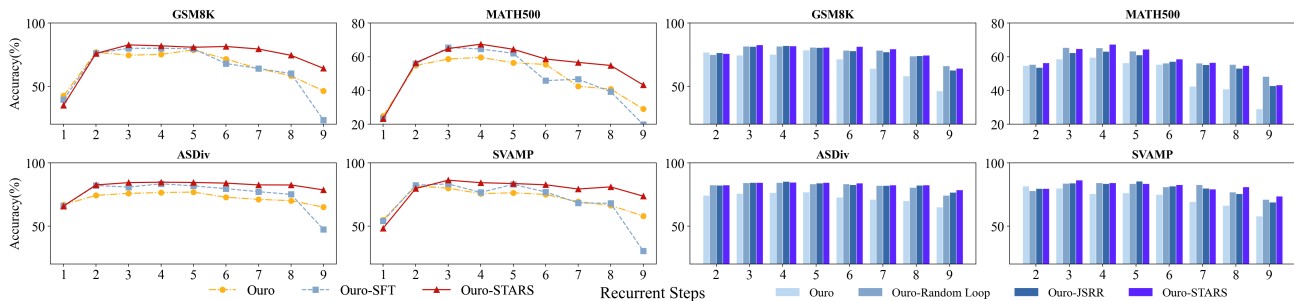

*Figure 4.* Comparative analysis of our method against baselines and its ablation variants across mathematical reasoning benchmarks. The left panel illustrates accuracy versus recurrent steps for Ouro, Ouro-SFT, and Ouro-STARS. The right panel details an ablation study evaluating the base Ouro model, Ouro with a Random Loop, Ouro with JSRR, and Ouro-STARS.

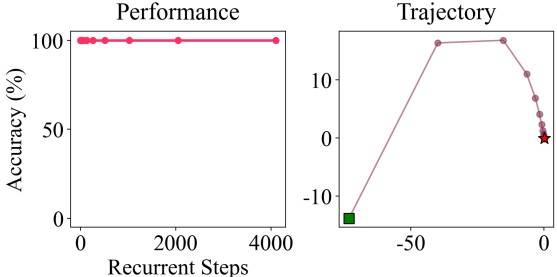

*Figure 5.* Performance and state evolution on the multi-digit addition task. The left panel displays the performance curve across recurrent steps, demonstrating the model's stability. The right panel illustrates the PCA-projected hidden state dynamics.

0.4, range $= [1, 16]$) and the weight $\lambda = 0.1$. The model was trained for one epoch across four NVIDIA A800 GPUs using AdamW (Loshchilov & Hutter, 2017) and a cosine learning rate scheduler starting at $1 \times 10^{-6}$. Subsequently, we use the lm_eval harness (Gao et al., 2024) to evaluate model's performance in a zero-shot setting across five mathematical benchmarks: GSM8K (Cobbe et al., 2021), MATH500 (Lightman et al., 2023), ASDiv (Miao et al., 2020), SVAMP (Patel et al., 2021), and AMC23 (Yang et al., 2025).

## 6.2. Main Results

**Multi-digits addition task.** As illustrated in Figure 5, with the integration of our proposed STARS, the model demonstrates notable robustness in performance. Regardless of the number of recurrent iterations, accuracy consistently remains at 100%, indicating that the model has stably learned the task. From a dynamical systems perspective, the latent states converge successfully to a stable fixed point with a relatively fast convergence rate. This ensures reliable reasoning even as the number of recurrent steps increases.

**Mathematical reasoning task.** Table 2 summarizes the performance of our framework in comparison to base Ouro-1.4B model and a standard Supervised Fine-Tuning (SFT) baseline across the five mathematical reasoning benchmarks.

We also compare with existing open-source small models. As anticipated and consistent with prior observations on recurrent models, both the base Ouro-1.4B and the SFT baseline exhibit significant performance degradation when the number of recurrent steps is extended beyond the nominal training depth (e.g., scaling from 4 to 8 recurrents). Specifically, the SFT baseline's average accuracy plummets from $70.46\%$ at 4 steps to a collapse at $52.97\%$ at 8 steps. In contrast, Ouro-1.4B-STARS (Ours) achieves the highest overall in-distribution performance, peaking at an average accuracy of $74.18\%$ at 4 recurrent steps. More critically, when subjected to significant depth-scaling to 8 recurrent steps, our framework maintains a remarkably robust average accuracy of $65.55\%$. As shown in the right panel of Figure 4, our method exhibits a slower decline in performance after reaching its peak as recurrent steps increase, demonstrating more stable scaling behavior.

## 6.3. Ablation Study

To thoroughly validate the effectiveness and quantify the individual contributions of each component in our proposed framework, we conduct a comprehensive ablation study. We evaluate performance across four representative mathematical reasoning benchmarks: GSM8K, MATH500, ASDiv, and SVAMP. Specifically, we compare the original Ouro-1.4B base model against three key variants: one integrated only with random loop sampling (Ouro-Random Loop), one with only JSRR (Ouro-JSRR), and the full STARS combining both strategies (Ouro-STARS). As shown in the right panel of Figure 4, the results demonstrate that STARS generally exhibits a slower decline in performance compared to the ablated variants, with each component, random loop sampling and JSRR, contributing to this delayed degradation. We provide additional analysis in Appendix C.

## 7. Conclusion

In this paper, we identify and diagnose the unreliable scaling behavior in current LoopLMs from a dynamic systems

perspective, attributing it to an inherent trade-off between reasoning effectiveness and latent trajectory stability. To resolve this, we propose STARS, a unified training framework that enforces asymptotic stability via Jacobian Spectral Radius Regularization alongside random loop sampling. Experimental results on both synthetic tasks and mathematical reasoning benchmarks demonstrate that STARS enables more reliable test-time scaling over greater recurrent depths compared to prior methods.

## Impact Statement

This paper presents work whose goal is to advance the field of Machine Learning. There are no potential societal consequences of our work which we feel must be specifically highlighted here. Our contributions are primarily methodological, focused on improving the stability and scaling behavior of a specific class of recurrent language model architectures for reasoning tasks. We do not introduce new applications or datasets, nor does our work directly address issues of fairness, safety, or bias in deployed systems. The proposed techniques are evaluated on standard academic benchmarks for mathematical reasoning.

## Acknowledge

This research was supported by the Jiangsu Science Foundation (BK20243012, BG2024036, BK20232003), Natural Science Foundation of China (62576162), and the Fundamental Research Funds for the Central Universities (022114380023).

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

## A. Limitations and Future Work

First, due to the computational resources limit, the empirical evaluation is concentrated on mathematical reasoning, leaving its generalizability to other complex reasoning domains (e.g., commonsense or strategic planning) an open question. Second, even on these tasks, while STARS prevents catastrophic collapse and enables scaling beyond the training horizon, performance does not always improve monotonically with more steps. This indicates that achieving fully reliable and predictable test-time scaling remains a challenge for the most difficult problems.

Future work will explore extending the STARS framework to a wider variety of reasoning and planning tasks, and investigating adaptive mechanisms to more robustly guide latent trajectories toward optimal, stable fixed points.

## B. Training Algorithm Details

---

**Algorithm 1** STARS Training for Looped Language Models with JSRR

---

**Input:** Dataset $\mathcal{D}$, Initial parameters $\theta$, Distribution $\mathcal{P}$, Regularization weight $\lambda$, Learning rate $\eta$, Power iteration steps $K$

**repeat**

    Sample a batch $\{\mathbf{x}^{(i)}, \mathbf{y}^{(i)}\}_{i=1}^{N} \sim \mathcal{D}$

    Sample loop depth $t \sim \mathcal{P}$

    **for** $i = 1$ **to** $N$ **do**

        *// Forward pass to loop depth $t$*

        $\mathbf{h}_i^{(0)} \leftarrow \text{prelude}(\mathbf{x}^{(i)})$

        $\mathbf{h}_i^{(t)} \leftarrow \underbrace{\Phi_\theta \circ \cdots \circ \Phi_\theta}_{t \text{ times}}(\mathbf{h}_i^{(0)})$

        *// Initialize random direction*

        $\mathbf{v}^{(i)} \sim \mathcal{N}(0, I)$

        $\mathbf{v}^{(i)} \leftarrow \mathbf{v}^{(i)} / \left( \|\mathbf{v}^{(i)}\|_2 + \epsilon \right)$

        *// Power iteration using Jacobian-vector products*

        **for** $k = 1$ **to** $K$ **do**

            $\mathbf{j}^{(i)} \leftarrow \text{JVP}\left( \Phi_\theta, \mathbf{h}_i^{(t)}, \mathbf{v}^{(i)} \right)$

            $\mathbf{v}^{(i)} \leftarrow \mathbf{j}^{(i)} / \left( \|\mathbf{j}^{(i)}\|_2 + \epsilon \right)$

        **end for**

        $\mathcal{L}_{\text{JSRR}}^{(i)} \leftarrow \|\mathbf{j}^{(i)}\|_2^2$

    **end for**

    *// Loss computation*

    $\mathcal{L}_{\text{SFT}}^{(t)} \leftarrow \frac{1}{N} \sum_{i=1}^{N} - \log p_\theta^{(t)}(\mathbf{y}^{(i)} \mid \mathbf{x}^{(i)})$

    $\mathcal{L}_{\text{JSRR}}^{(t)} \leftarrow \frac{1}{N} \sum_{i=1}^{N} \mathcal{L}_{\text{JSRR}}^{(i)}$

    $\mathcal{L} \leftarrow \mathcal{L}_{\text{SFT}}^{(t)} + \lambda \mathcal{L}_{\text{JSRR}}^{(t)}$

    *// Optimization step*

    $\theta \leftarrow \theta - \eta \nabla_\theta \mathcal{L}$

**until** convergence

---

## C. More Results

**Random loop sampling on PreNorm and PostNorm** We supplement the experimental results of random loop sampling for PreNorm with LN and PostNorm with LN under three distributions (Log-Normal, Poisson, and Uniform), each with two configurations, as shown in Figure C and Table 3.

**Hyperparameter analysis.** We conducted a hyperparameter analysis by evaluating the regularization weight $\lambda$ on multi-digit addition tasks, with the results illustrated in 7. This investigation involved plotting the performance curve and the PCA-projected trajectory for a range of $\lambda$ values. The weight $\lambda$ directly influences the strength of the JSRR constraint, which, in turn, is reflected in the resulting latent dynamics. While our method demonstrates that different weights lead to the model converging to attractor, as shown in the trajectory plots, if the weight $\lambda$ is too large (e.g. 0.15 and 0.20), the model

*Figure 6.* The results with the random loop strategy across distributions and parameter sets for PreNorm with LN and PostNorm with LN.

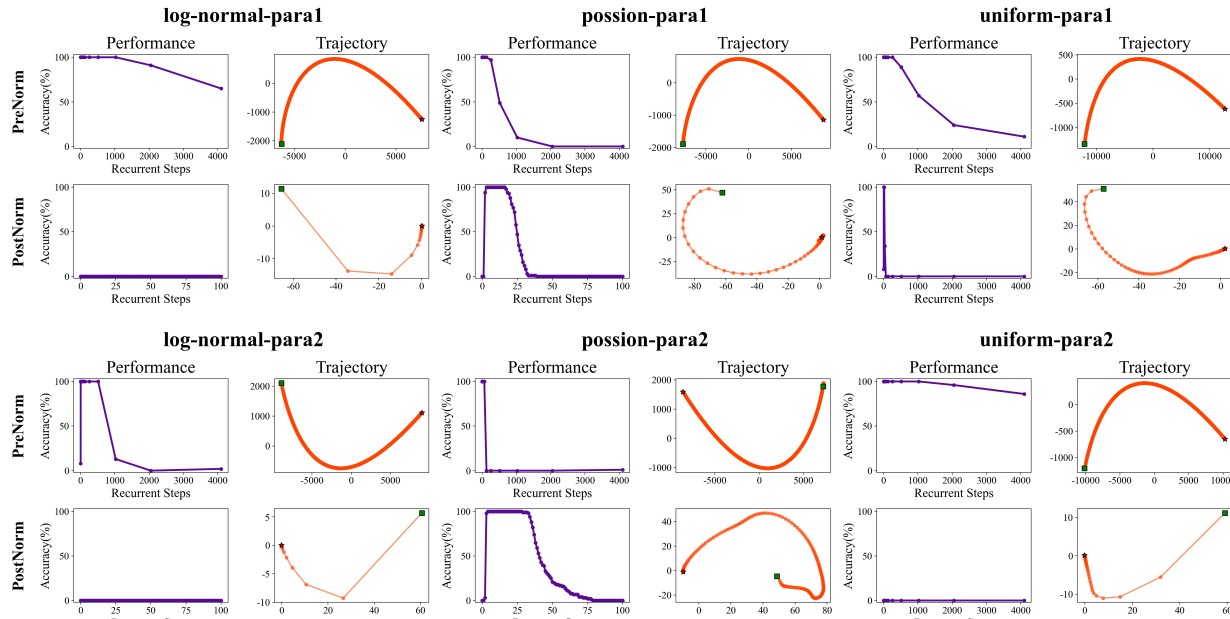

*Table 3.* Hyperparameter settings for random loop distributions with PreNorm and PostNorm. Range indicates the clipping bounds $[\min, \max]$.

| Distribution | Set 1 Configuration | Set 2 Configuration |
|---|---|---|
| Log-Normal | $\mu = 2.62, \sigma = 0.60$ range: $[1, 40]$ | $\mu = 3.2, \sigma = 0.45$ range: $[1, 80]$ |
| Poisson | $\lambda = 5$ range: $[1, 30]$ | $\lambda = 10$ range: $[1, 30]$ |
| Uniform | range: $[1, 10]$ | range: $[1, 30]$ |

struggles to learn to solve the original multi-digit task. This is evident in the corresponding performance curves, which show a significant drop in accuracy. Therefore, although our method exhibits a degree of robustness, the regularization weight should not be excessively large, as it can impede the model's effective learning capabilities.

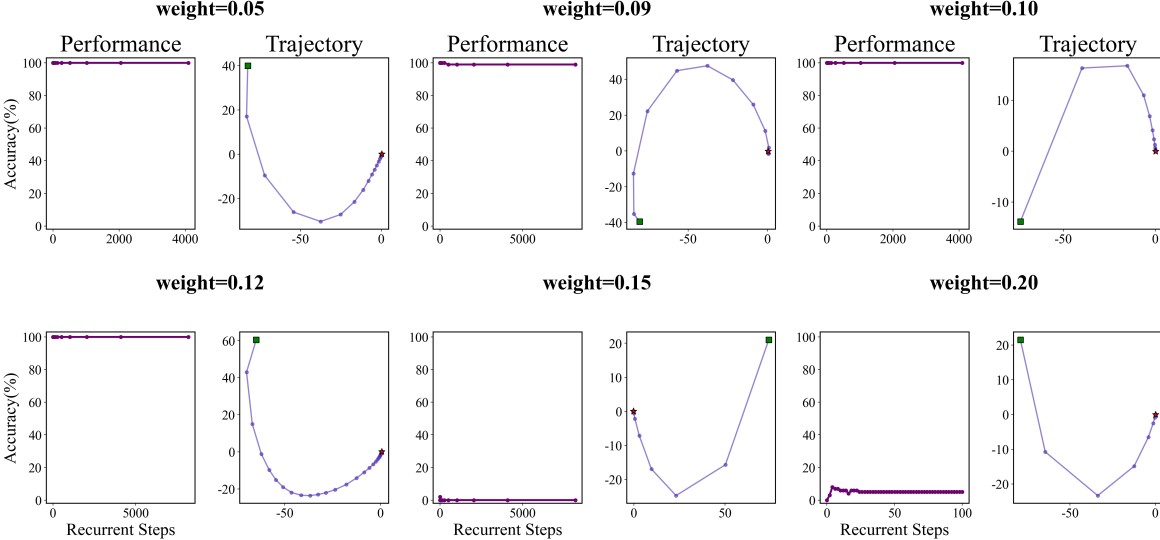

*Figure 7.* Hyperparameter analysis on the multi-digits addition task.

**Efficiency analysis during training phase.** We conducted an efficiency analysis during the training phase for these four types, with Ouro-SFT as the baseline. The results are shown in Table 4. From the table, it can be observed that the efficiency of Ouro-Random Loop, Ouro-JSRR, and Ouro-STARS during the training phase is 1.4976, 1.5317, and 2.0439, respectively, relative to Ouro-SFT.

*Table 4.* Efficiency analysis during training phase. The values are presented relative to Ouro-SFT as the base.

| Model | Ouro-SFT | Ouro-Random Loop | Ouro-JSRR | Ouro-STARS |
|---|---|---|---|---|
| **Relative Value** | 1 | 1.4976 | 1.5317 | 2.0439 |

**Comparative Analysis and Ablation Study on AMC23.** In addition to the mathematical reasoning benchmark analysis presented in the main text (Figure 4), we further conducted a comparative analysis of our method against baselines and its ablation variants on the AMC23 dataset shown in Figure 8.

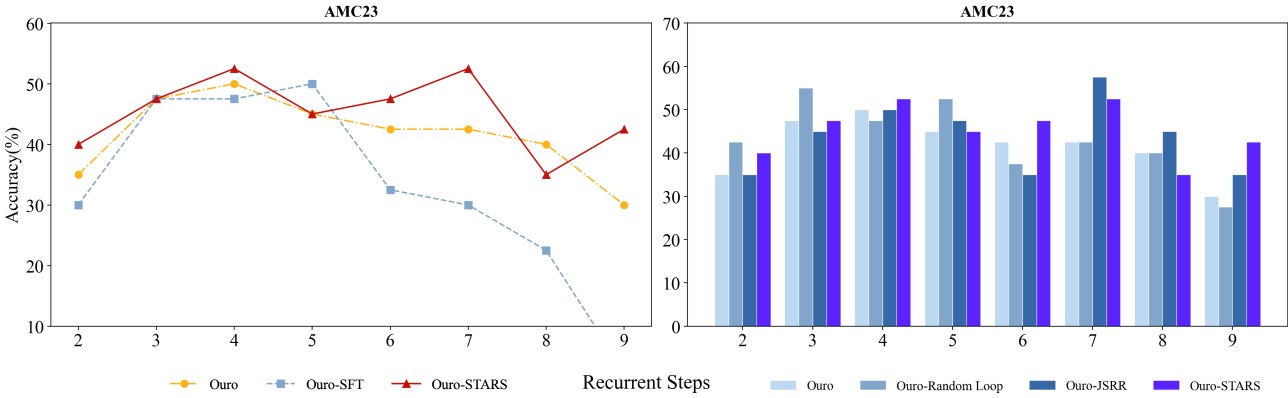

*Figure 8.* Comparative analysis of our method against baselines and its ablation variants on AMC23.

