# OpenReview forum: "Stabilizing Recurrent Dynamics for Test-Time Scalable Latent Reasoning in Looped Language Models"
_ICML.cc/2026/Conference — ICML 2026 regular_

### Official Review · Reviewer_qf2p · 2026-02-20

**Soundness:** 4
**Presentation:** 3
**Significance:** 3
**Originality:** 3
**Overall Recommendation:** 5
**Confidence:** 4

**Summary:**

The paper presents a thorough study of recurrent depth transformer on their performance and stability. The paper first studies on synthetic 4-digit addition task and observes a tradeoff between performance and stability depending on the normalization position. The paper further shows that other techniques such as L2 regularization does not significantly change it. Then the authors use a principled approach called Jacobian Spectral Radius Regularization to mitigate the stability issues. On synthetic and real-world math datasets, the proposed fix shows effectiveness.

**Compliance With Llm Reviewing Policy:**

Affirmed.

**Final Justification:**

I believe the author has done a great job in studying recursive depth of transformers and the insights are beneficial to the community.

**Key Questions For Authors:**

For math reasoning task, why does it use a different random loop sampling range than the addition task?

What might be the reason for performance degradation for math reasoning unlike the addition task when the recurrent depth increases?

Why is recurrent depth of 8 called extrapolation where the range is between 1 and 16?

**Limitations:**

Yes

**Strengths And Weaknesses:**

The paper is very well-written and I have no problem understanding the results and why these experiments are conducted. The studies of normalization positions are particularly sound. The JSSR fix is well-motivated as well. Given the significant performance and stability improvement, it is also significant contribution.

The main weakness I believe are imprecise languages and typo. For example “an effective” -> “an effective”, “produc” -> “product”.

---

> ### Author Rebuttal · Authors · 2026-03-30
>
> We thank the reviewer for the thoughtful comments. We address the concerns below.
>
> 1. On imprecise languages and typo. (Weakness)
>
>     We sincerely thank the reviewer for the careful reading and for pointing out these errors. We have carefully proofread the text and will correct these errors in the final version of the paper.
>
> 2. On the different random loop sampling range between math reasoning task and the addition task. (Q1)
>
>      The choice of the sampling range for the math reasoning task involves a trade-off between stability optimization and maintaining compatibility with the pre-trained model. First, since the Ouro-1.4B model was pre-trained with a limited number of iterations, applying a significantly larger range during post-training creates a substantial distributional shift, which can destabilize the model's performance. Our range represents a balance between stability regularization and preserving the pre-trained behavior. Second, given the scale of the 1.4B parameter model and the complexity of math reasoning, a broader sampling range imposes significant memory and compute overhead.
>
>
> 3. On the possible reason for performance degradation for math reasoning. (Q2)
>
>     We agree that our method does not fully eliminate performance degradation at sufficiently long horizons for complex tasks. A likely reason for this is that, in the Ouro experiments, the proposed stability objective is applied only during the SFT phase, while the original pre-training was performed without such stability constraints. As a result, the model is not fully optimized for the desired stability properties throughout its entire training pipeline. We believe that achieving stronger long-horizon mitigation may require incorporating this objective during the pre-training stage, similar to our approach in the controlled addition experiments. Nevertheless, our method still provides a substantial improvement in robustness when increasing the recurrence depth, and we will include this discussion in the revised manuscript to better address the limitations and potential directions for future work.
>
>
> 4. On the characterization of a recurrent depth of 8 as "extrapolation". (Q3)
>
>     We thank the reviewer for pointing out this inconsistency. We acknowledge that calling a recurrence depth of 8 "extrapolation" is imprecise given that our training range spans from 1 to 16. This was a misstatement in our manuscript. Our objective is to ensure that the model’s performance remains stable or continues to improve as the recurrence depth increases within the training range, rather than to test its out-of-distribution generalization. We will correct this terminology in the revised version of the paper to ensure a more accurate description of our experimental results.

---

> > ### Author Rebuttal · Reviewer_qf2p · 2026-04-01
> >
> > I am happy to keep my positive rating.

---

### Official Review · Reviewer_ydA1 · 2026-02-26

**Soundness:** 3
**Presentation:** 3
**Significance:** 3
**Originality:** 3
**Overall Recommendation:** 4
**Confidence:** 4

**Summary:**

This paper explains why Looped Language Models often do not improve with deeper test-time recurrence: performance can peak and then collapse due to unstable latent dynamics. It shows normalization placement largely determines two failure modes. To stabilize recurrent scaling, it proposes STARS, combining random loop-depth sampling with Jacobian Spectral Radius Regularization to encourage locally stable fixed points, using efficient JVP-based estimation.

**Compliance With Llm Reviewing Policy:**

Affirmed.

**Final Justification:**

The author addressed my concern. Thanks

**Key Questions For Authors:**

1. Have you considered combining STARS with the Ouro early-exit gate, training it with an explicit benefit–cost objective (performance gain vs. extra recurrent steps) so the model can stop when returns diminish, improving deployability over fixed deep loops?

2. Do you have results for Ouro-2.6B trained with STARS? Specifically, how does STARS affect its peak accuracy and stability under deeper recurrent inference.

**Limitations:**

Yes

**Strengths And Weaknesses:**

**Strengths**

1. The authors systematically vary normalization choices and placements, and clearly demonstrate that normalization placement is the dominant factor shaping recurrent dynamics and failure modes.

2. The evaluation protocol is well-aligned with the stated objective of recurrent test-time scaling, by training with a fixed recurrent depth and explicitly probing generalization under deeper inference-time recurrence.

3. The paper includes multiple competitive control strategies and reports meaningful negative results, which strengthens the empirical narrative and helps isolate the contribution of STARS.


**Weaknesses**
1. While the use of JSRR is a reasonable design choice to reduce overhead, the paper currently provides mostly methodological justification rather than hard efficiency measurements. To support the scalability claim, please report concrete training and inference costs.

2. The PCA trajectory visualizations are intuitive, but they are not a rigorous or easily reproducible measure of dynamical stability.  The empirical case would be stronger with numerical stability metrics, such as the growth rate of (|h^{(t)}|) as a function of recurrent depth (t).

---

> ### Author Rebuttal · Authors · 2026-03-30
>
> We thank the reviewer for the careful reading and the constructive feedback. We address the comments below.
>
> 1. On concrete efficiency measurements (W1)
>
>     Thank you for this suggestion. We agree that concrete cost reporting is important for supporting the scalability claim. We have in fact reported the training cost in Appendix Table 4, where the relative training efficiency is measured against Ouro-SFT: Ouro-Random Loop = 1.4976, Ouro-JSRR = 1.5317, and Ouro-STARS =2.0439.  Due to the addition of new training targets, there will inevitably be a sacrifice in efficiency.
>     For inference, our method does not modify the inference procedure itself, so cost remains the same as in the underlying LoopLMs.
>
> 2. On PCA trajectories not being a rigorous numerical stability metric. (W2)
>
>     We agree with the reviewer that PCA trajectories are primarily an intuitive visualization rather than a rigorous standalone stability measure. Their role in the current paper is to make the latent dynamics visually interpretable under deeper recurrence.   We provide the results of hidden-state norm (|h^{(t)}|) as a function of recurrent depth. The numerical results are fully consistent with the PCA trends. In PreNorm, the hidden-state norm grows rapidly with depth (t): before training, it increases from 0.64 at (t=0) to 39.55 at (t=8) and 126.81 at (t=23); after training (without random loop or regularization), it grows even faster, from 0.76 at (t=0) to 54.89 at (t=8) and 135.64 at (t=23). By contrast, in PostNorm, the hidden-state norm remains tightly bounded: before training, it reaches 22.63 at (t=1) and then stays essentially constant (22.6273 at (t=23)); after training, it still remains in a very narrow range, from 22.59 at (t=1) to 22.61 at (t=23). These results provide a direct numerical characterization of the two regimes highlighted by the PCA plots: PreNorm exhibits unbounded norm growth with recurrent depth, whereas PostNorm stays bounded but tends to converge to ineffective states. We will add hidden-state norm curves and corresponding growth statistics to the revised paper.
>
> 3. On combining STARS with the Ouro early-exit gate. (Q1)
>
>     This is an excellent suggestion. We agree that combining STARS with an early-exit mechanism could improve deployability by allowing the model to stop when the marginal benefit of additional recurrent steps diminishes. However, this extension is beyond the scope of the current paper. Our goal here is to isolate and validate the core contribution of STARS from the perspective of dynamical stability. We will highlight adaptive stopping as an important direction for future work.
>
> 4. On results for Ouro-2.6B. (Q2)
>
>     We agree that evaluating STARS on Ouro-2.6B would be valuable for assessing both peak accuracy and stability under deeper recurrent inference. Unfortunately, due to rebuttal-time and compute constraints, we are unable to provide these results at this stage. We will note this more clearly as future work. At present, our main claim is supported by the controlled addition experiments and the Ouro-1.4B study, where STARS improves both peak accuracy and robustness when increasing loops.

---

> > ### Author Rebuttal · Reviewer_ydA1 · 2026-04-01
> >
> > Thanks. I am happy to keep my rating.

---

### Official Review · Reviewer_4tDX · 2026-03-02

**Soundness:** 2
**Presentation:** 3
**Significance:** 2
**Originality:** 2
**Overall Recommendation:** 4
**Confidence:** 3

**Summary:**

This paper tackles why "looped" AI models—which reuse the same layers to "think" deeper—often crash or lose accuracy when you let them run for extra iterations during inference. By analyzing these models as dynamical systems, the authors discovered a fundamental trade-off where standard designs either lead to internal logic that explodes out of control (Pre-Norm) or stays too shallow to solve hard problems (Post-Norm). To fix this, they introduced STARS, a training framework that uses a math trick called Jacobian Spectral Radius Regularization to force the model's internal "thoughts" to settle into stable, reliable states rather than drifting into chaos. In tests on math tasks like GSM8K, this approach not only boosted peak performance by 4% but also made the model much tougher, cutting performance collapses by more than half when the reasoning depth was doubled.

**Compliance With Llm Reviewing Policy:**

Affirmed.

**Final Justification:**

i raise my score

**Key Questions For Authors:**

can the author give justification about the assumption of Lyapunov Linearization Theorem?

can the author give approxiamtion error about the JSSR?

could you please provide a head-to-head comparison with large non-looped models?

can you give further explanation about the performance degradation as the steps increase? I think if the loop architecture reach the fixed points the performance should reach a pleatue instead of degrading.

**Limitations:**

see above

**Strengths And Weaknesses:**

# strength
The paper provides a novel diagnostic study of Looped Language Models through the lens of dynamical systems, identifying a critical trade-off between effectiveness and stability in latent trajectories.

# weakness
**Theoretical Assumptions on Lyapunov Linearization:** The framework relies heavily on the Lyapunov Linearization Theorem to justify local stability through the Jacobian spectral radius. However, the paper lacks a rigorous verification of whether the latent reasoning process satisfies the necessary conditions for this theorem (e.g., the existence of a hyperbolic fixed point or the differentiability requirements of the nonlinear map $\Phi_{\theta}$ in the context of high-dimensional Transformer states).

**Lack of Error Analysis for JSRR Approximation:** JSRR is implemented as an approximation due to the prohibitive cost of full Jacobian computation. The authors do not provide a theoretical bound or empirical validation of the approximation error between the one-step power iteration estimate and the true spectral radius. A "toy example" where the full Jacobian is computable would have been beneficial to justify this approximation. For example, a two-layer MLP is not as complicated as the transformer, and you can compute the jacobian to give some insights about the approximation

**Incomplete Efficiency Benchmarking:** While the paper argues that LoopLMs offer improved computational efficiency by avoiding longer context windows, it fails to compare the increased inference cost of multiple recurrent steps against larger, non-recurrent models (e.g., 7B or 14B parameters). Without a "performance-per-FLOP" comparison against standard high-capacity models, the practical utility of depth-recurrence remains unproven.

**Non-Monotonic Performance:** As noted in the limitations, performance does not always improve monotonically with additional steps. This suggests that while STARS mitigates "collapse," it hasn't yet achieved the ideal behavior where more compute always equals better reasoning. I think the performance improvement from the looped structure is quite limited, perhaps because of the paradigm itself. So my concerns is that instead of increasing looped steps of the transformer why not increase the layer of the model.

---

> ### Author Rebuttal · Authors · 2026-03-30
>
> We thank the reviewer for the thoughtful comments. We address the concerns below.
>
> 1. On the assumptions behind the Lyapunov Linearization Theorem. (W1 & Q1)
>
>     We acknowledge that rigorously verifying whether the latent reasoning process in a high-dimensional Transformer satisfies the assumptions of the Lyapunov Linearization Theorem is challenging, as different parameterizations can yield different nonlinear dynamics. However, given the large number of learnable parameters, we believe these assumptions are practically reasonable, as Transformers are differentiable systems, and similar local stability analyses appear in prior work such as DEQ [1,2]. Our aim is not to provide a complete global proof for the learned reasoning dynamics, but to use the theorem as a principled local motivation for regularizing the Jacobian spectral radius.
>
> 2. On the approximation error of JSRR. (W2 & Q2)
>
>     We thank the reviewer for this valuable suggestion. To better understand the approximation error of JSRR, we performed an additional empirical study on the addition task (fix loop=10), where we compare the one-step JSRR estimate with a near-converged power-iteration estimate used as a reference spectral radius. Our results confirm that JSRR is an approximation and exhibits a bias in magnitude (about 0.2-0.3). However, the optimization behavior is still well preserved: when we optimize the JSRR objective, the reference spectral radius also decreases. Concretely, the JSRR estimate decreases from an initial value of 0.7025 to 0.0003 at convergence, while the reference spectral radius decreases from 1.0397 to 0.23, which is well below 1.0. Moreover, the gradients of our method and the reference loss remain highly aligned during training (gradient cosine similarity: 0.99). These observations suggest that although JSRR is not an exact estimator, it captures a useful training direction for reducing the true spectral radius. We agree that this point deserves clearer discussion in the paper. In the revised version, we will add these empirical results and expand the discussion on the approximation behavior of JSRR.
>
> 3. On incomplete efficiency and comparison with non-looped models. (W3 & Q3)
>
>     We thank the reviewer for raising this important point. We agree that a compute-normalized comparison between LoopLM and larger non-recurrent models is valuable. However, the goal of this paper is not to provide a comprehensive architectural comparison between LoopLM and dense LLMs, but to study the stability and effectiveness of an existing LoopLM (based on Ouro [3]) under test-time scaling. For this reason, we retain the original Ouro architecture and inference mechanism, rather than introducing a new setup for a fair dense-vs-looped comparison. The non-recurrent results in Table 2 are included mainly for reference. Meanwhile, the original Ouro work [3] already provides relevant context, including comparisons with larger standard Transformers and evidence from Figure 6 on iso-parameter and iso-FLOP baselines in synthetic tasks. We will clarify in the revision that our work should be understood as a study of the test-time scaling behavior of LoopLM, rather than a definitive claim of superiority over non-recurrent models.
>
> 4. On performance degradation. (W4 &Q4)
>
>    We agree that fully achieving the ideal scaling law that more recurrent compute always improves reasoning remains an open problem for current large LoopLMs. Our goal here is not to claim that this problem is solved, but to make a concrete empirical step toward it by diagnosing the failure mode and substantially mitigating collapse. In Ouro, our method is applied only during SFT, while pretraining is conducted without the proposed stability objective. We therefore believe the model is not yet trained end-to-end to reliably converge to good fixed points, which likely explains why performance can still degrade rather than plateau at long horizons. By contrast, in the controlled addition setting, where the stability objective is used throughout training, the behavior is much closer to the desired trend and remains stable under increased recurrence. Regarding why not directly increase Transformer depth, we view these as two distinct scaling axes: loop steps test whether repeated application of a shared transformation can provide more useful computation at fixed parameters. Prior work [4,5] suggests that looped architectures are worth studying as a distinct direction for iterative refinement, input-adaptive compute allocation, and potentially different expressive properties. We will make this positioning clearer in the revision.
>
> [1] Deep Equilibrium Models. NeurIPS 2019.
>
> [2] Stabilizing Equilibrium Models by Jacobian Regularization. ICML 2021.
>
> [3] Scaling Latent Reasoning via Looped Language Models. Arxiv 2025.
>
> [4] Looped Transformers are Better at Learning Learning Algorithms. ICLR 2024.
>
> [5] On Expressive Power of Looped Transformers. ICML 2025.

---

> > ### Author Rebuttal · Reviewer_4tDX · 2026-04-01
> >
> > i raise the score

---

### Official Review · Reviewer_Yx4o · 2026-03-05

**Soundness:** 3
**Presentation:** 4
**Significance:** 3
**Originality:** 3
**Overall Recommendation:** 5
**Confidence:** 4

**Summary:**

This paper addresses the issue of unstable dynamics in looped transformer language models. These models offer the promise of increased expressiveness through deeper looping, but empirical investigations show that performance degrades significantly after a certain reasoning depth has been reached. This phenomenon is especially pronounced in Ouro models, currently the only LoopLM model which competes with open-source LLMs. The authors run a detailed empirical analysis of the performance of looped LLMs on the task of 4-digit addition, finding that looped transformer models seem to trade off norm stability and performance with increased reasoning depth. Further analyses take into account different recurrence depth sampling strategies and the effect of an L2 regularization objective aimed to constrain the divergence between subsequent recurrent representations. Finally, the authors propose a method based on a Monte Carlo approximation of the spectral radius of the recurrent map, and demonstrate that incorporating the spectral radius in the optimization objective enables looped transformer models to retain accuracy as well as maintain controlled state norms under increased reasoning depths compared to the baselines.

**Compliance With Llm Reviewing Policy:**

Affirmed.

**Final Justification:**

I happily maintain my positive score.

**Key Questions For Authors:**

**Questions**
- Do you normalize the embeddings in the models you tested in Figure 2? Does Ouro normalize the embeddings? I am asking about it as I am surprised by the very large magnitude of the network output already at one recurrent step.
- Could you show data on the estimated spectral radius of the recurrent map in different scenarios? This should help ground the proposed method in empirical observations.

**Limitations:**

yes

**Strengths And Weaknesses:**

**Strengths**
- The presentation is very well put-together, I rarely have the pleasure of reading such a well-planned research text. The text flows smoothly, and information is presented in an order which significantly aids understanding.
- The observations in Figure 2 are quite interesting. The unstable systems can maintain performance at significantly deeper recurrences than the stable systems. This indeed is an interesting trade-off uncovered here.
- The stabilization method is clean and clearly effective. Based on the results from Table 2, accuracy is retained to a higher degree with more loop iterations than without the proposed regularization method, and we can even observe accuracy gains at recurrence depth 4 as compared to Ouro-1.4B.


**Weaknesses**
- 4.2.2 seems like a bit of a redundant section. Did we a-priori have any reason to believe that the prelude and coda layers would alter the system dynamics?
- I think there's something wrong with the following statement in section 4.2.1, "This creates a positive feedback loop where the state magnitude amplifies exponentially with iterations, eventually diverging from the functional data manifold.". The exponential increase only becomes relevant once we discuss the discrete system perspective and the spectral radius in section 5; it should not be a consequence of the residual accumulation, which should incur a linear additive increase.
- The method still does not prevent a performance decrease when the number of recurrence iterations is longer than a certain threshold (e.g. 4 in case of Table 2).
- The estimated spectral radius is never explicitly shown. Is it really larger than 1 in most unstable cases?

**Presentation Suggestions**

- In section 4.3.2., the text leads us to believe that the proposed L2 normalization yields no positive benefits; however, the best results in Figure 3 with an externally normalized system are achieved with the L2 regularization (bottom right quadrant, post-sandwich log-normal-para2 and poisson-para2).

---

> ### Author Rebuttal · Authors · 2026-03-30
>
> We sincerely thank the reviewer for the careful reading, the positive assessment of the paper and the constructive suggestions. We address the concerns point by point below.
>
> 1. On Section 4.2.2 (Prelude/Coda) seeming redundant. (W1)
>
>     We want to clarify that our motivation for including this analysis was not that we had a strong  priori claim that the prelude and coda layers must alter the system dynamics, but rather that prior empirical work [1,2] had already used such design variations, while the mechanism behind their effect remained unclear. We therefore included this section to better disentangle where stability behavior may enter the looped architecture. And actually we find that the inclusion of Prelude and Coda layers does not alter the system's dynamical tendencies. We consider this a no trivial result.
>
> 2. On the wording in Section 4.2.1 regarding exponential amplification. (W2)
>
>     We agree that the current wording is imprecise. Residual accumulation alone suggests a linear additive effect. We will revise the sentence accordingly and make the mechanism more precise.
> 3. On performance still degrading beyond a certain recurrence depth. (W3)
>
>     We agree that our method does not fully eliminate degradation at sufficiently long horizons for complex tasks. A likely reason is that, in the Ouro experiments, the proposed objective is applied only during SFT, while pretraining is performed without this stability objective. As a result, the model is not optimized for the desired stability properties throughout the full training pipeline. We therefore believe stronger long-horizon mitigation may require incorporating the objective already during pretraining, as in our controlled addition experiments. That said, our method already substantially improves robustness when increasing depth.
>
> 4. On the estimated spectral radius not being shown explicitly. (W4 & Q2)
>
>     We agree that showing it explicitly would strengthen the paper. We have now examined the estimated spectral radius in the addition experiments under typical settings. The results support our proposed mechanism: in unstable settings, the estimated spectral radius is generally above 1 across a broad recurrence range, while under our method it decreases below 1 as recurrence grows. These results give direct empirical support to the role of JSRR: random loop training alone slightly reduces the spectral radius but does not fundamentally change the unstable regime, whereas our method drives it below 1 from step 20 onward, consistent with improved stability and scaling.
>
>     | Recurrent steps | Sandwich-PostNorm (before training) | PreNorm (before training) | PreNorm (after training) | PreNorm + random loop (after training) | Sandwich-PostNorm + STARS |
>     |---|---:|---:|---:|---:|---:|
>     | 5  | 1.059 | 1.251 | 1.133 | 1.206 | 1.256 |
>     | 10 | 1.019 | 1.104 | 1.068 | 1.062 | 1.061 |
>     | 15 | 1.006 | 1.055 | 1.044 | 1.032 | 1.007 |
>     | 20 | 1.005 | 1.035 | 1.029 | 1.021 | **0.996** |
>     | 25 | 1.007 | 1.024 | 1.021 | 1.014 | **0.993** |
>     | 30 | 1.008 | 1.017 | 1.014 | 1.011 | **0.991** |
>     | 35 | 1.009 | 1.013 | 1.011 | 1.008 | **0.994** |
>     | 40 | 1.009 | 1.011 | 1.008 | 1.006 | **0.989**|
>
>
> 5. On the interpretation of Figure 3 and L2 regularization. (Presentation Suggestion)
>
>     We would like to clarify that the configurations in the bottom-right quadrant of Figure 3 correspond to the random loop strategy across distributions and parameter sets, and do not include the proposed L2 regularization. The results for L2 regularization are shown in the left panel of Figure 3, where its performance is consistently weak. Therefore, our interpretation remains that L2 regularization does not provide clear positive benefits in our experiments. We will revise Section 4.3.2 and the figure description to make this distinction explicit.
>
> 6. On whether embeddings / hidden states are normalized in Figure 2 and in Ouro. (Q1)
>     Our goals is precisely to study the effect of normalization placement on the recurrent dynamics. For the post-norm and post-sandwich variants, normalization can be seen as applied after each recurrence step by design. As we show in the addition experiments, post-recurrence normalization plays an important role in stabilizing the dynamics. Accordingly, in Ouro, normalization is also applied after each loop iteration; in fact, this is already part of the Ouro architecture itself. We will clarify this setup in the revision.
>
> [1] Scaling up Test-Time Compute with Latent Reasoning: A Recurrent Depth Approach. ICML 2025.
>
> [2] Teaching Pretrained Language Models to Think Deeper with Retrofitted Recurrence. Arxiv 2025.

---

> > ### Author Rebuttal · Reviewer_Yx4o · 2026-04-04
> >
> > I will happily maintain my positive score.

---

### Decision · Program_Chairs · 2026-04-30

**Decision:**

Accept (regular)

**Comment:**

Reviewers found this paper to be clearly written, well motivated, and empirically strong.  They unanimously advocate for acceptance.